# Why Does *Phlebiopsis gigantea* not Always Inhibit Root and Butt Rot in Conifers?

**Anna Żółciak** [1,*] , **Katarzyna Sikora** [1] , **Marta Wrzosek** [2], **Marta Damszel** [3] and **Zbigniew Sierota** [4]

1 Department of Forest Protection, Forest Research Institute in Sękocin Stary, Braci Leśnej 3, 05-090 Raszyn, Poland; K.Sikora@ibles.waw.pl

2 Department of Molecular Phylogenetics and Evolution, University of Warsaw, Żwirki i Wigury 101, 02-089 Warsaw, Poland; martawrzosek@gmail.com

3 Department of Entomology, Phytopathology and Molecular Diagnostics, Faculty of Environmental Management and Agriculture, University of Warmia and Mazury in Olsztyn, Prawocheńskiego 17, 10-721 Olsztyn, Poland; marta.damszel@uwm.edu.pl

4 Department of Forestry and Forest Ecology, Faculty of Environmental Management and Agriculture, University of Warmia and Mazury in Olsztyn, Pl. Łódzki 2, 10-727 Olsztyn, Poland; zbigniew.sierota@uwm.edu.pl

* Correspondence: a.zolciak@ibles.waw.pl; Tel.: +48-22-7153-822

**Abstract:** This review aims to identify possible causes of differing effectiveness of artificial biological control of *Heterobasidion* root rot by the saprotrophic fungus *Phlebiopsis gigantea*. We describe published information in terms of pathogen–competitor relationships and the impact of environmental and genetic factors. We also revisit data from original research performed in recent years at the Forest Research Institute in Poland. We hypothesized that, in many cases, competition in roots and stumps of coniferous trees between the necrotrophic *Heterobasidion* spp. and the introduced saprotroph, *Phlebiopsis gigantea*, is affected by growth characteristics and enzymatic activity of the fungi, the characteristics of the wood, and environmental conditions. We concluded that both wood traits and fungal enzymatic activity during wood decay in roots and stumps, and the richness of the fungal biota, may limit biological control of root rot. In addition, we identify the need for research on new formulations and isolates of the fungal competitor, *Phlebiopsis gigantea*, as well as on approaches for accurately identifying the infectious threat from pathogens.

**Keywords:** competition; *Heterobasidion annosum*; *Heterobasidion parviporum*; mycelium growth; wood decay

## 1. Introduction

The necrotrophic fungi *Heterobasidion* spp. cause serious losses in coniferous forests in the Northern Hemisphere [1–6]. The problem is well known, with nearly 600 papers published by 1971 describing this issue in detail [7]. In more recent years, molecular methods have been used to describe both the genetic background and physiology of the infection process [8]. These studies have explained the different biological mechanisms governing parasitic interactions of both described pathogens with Scots pine (*Pinus sylvestris* L.) and Norway spruce (*Picea abies* (L.) H. Karst.) (Figure 1). *Heterobasidion annosum* (Fr.) Bref. develops in pine in the outer part of the roots (phloem, cambium) and *H. parviporum* (Niemelä and Korhonen) develops in spruce generally in the inner parts (heartwood). These differences should improve our understanding of the methods for prevention, dilution of breeding loss, and direct reduction of pathogen inoculum in roots and stumps (Figure 1) [9–12]. Despite some achievements in developing methods of monitoring the occurrence of this pathogen, some questions concerning its

genetics, biology and factors of pathogenicity remain unanswered. In particular, there are questions concerning methods of selecting trees resistant to the pathogen and factors influencing success of the biological prevention of infection using the competitive fungus *Phlebiopsis gigantea* (Fr.) Jülich, which itself is unable to infect living trees [13]. Infections by *Heterobasidion* spp. cause financial and breeding losses and so remain as scientific and economic issues and the subject of ongoing investigation (Figure 2) [14–16].

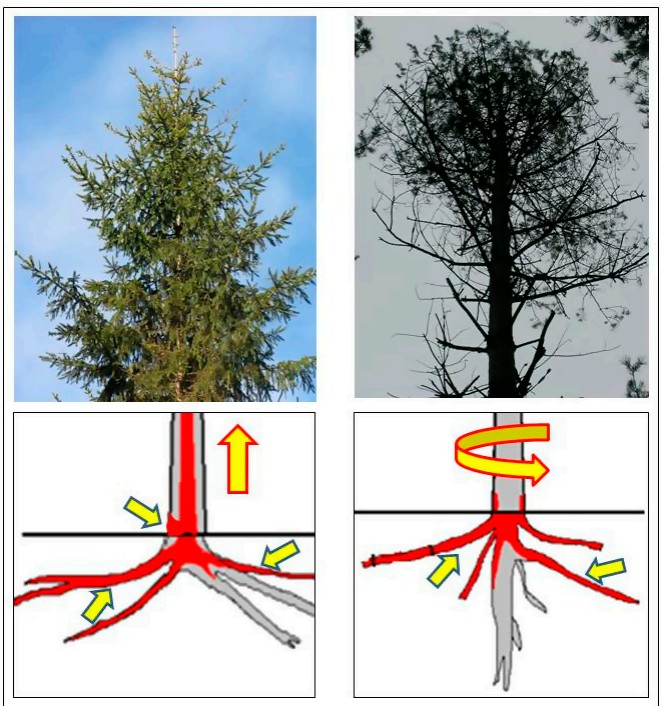

**Figure 1.** Differences in symptoms in the crowns of Norway spruce (left, no symptoms) and Scots pine (right, crown thinning, dying tree), with diagrams of *Heterobasidion* spp. mycelium spread in the affected trees (arrows).

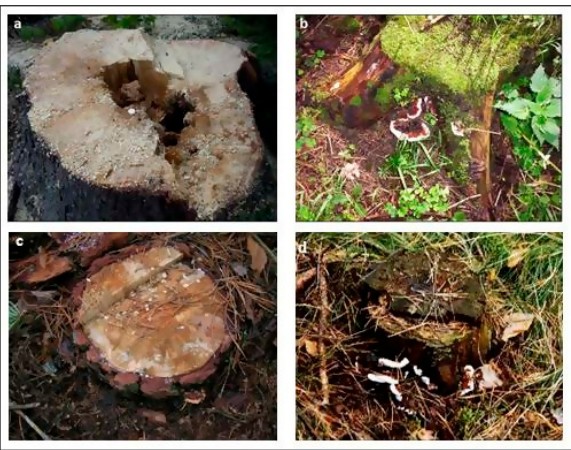

**Figure 2.** Wood decay (left) and sporocarps (right) in Norway spruce stumps infected with *H. parviporum* (upper (**a**,**b**)) and Scots pine stumps infected with *H. annosum* (lower (**c**,**d**)).

The elimination of *Heterobasidion* spp. in stumps with intact root systems using a natural competitor, the competitor *P. gigantea*, remains the only confirmed and effective method of reducing pathogen infection. Commercial preparations containing *P. gigantea* spores have been used in Europe for years, particularly in Scandinavia.

They are used to protect both pine and Norway spruce stands against root rot caused by *Heterobasidion* spp. [14]. Since 2015 they have been marketed in Poland as Rotstop WP [17–19]. *P. gigantea* is a proven competitor of several fungi in vitro. Hyphae of this fungus antagonize on contact the hyphae of other fungi, including *Heterobasidion spp.*, a phenomenon termed hyphal interference. Any hypha of *H. annosum* (Fr.) Bref. making contact with a hypha of *P. gigantea* shows rapid, localized disruption. The protoplasm becomes disorganized and membrane integrity is reduced [20,21]

In recent years, however, the effectiveness of this biological control method has been contradicted by research indicating potential limitations in its antagonistic effects under natural conditions. These limitations indicate the need to determine the conditions for achieving full, long-term effectiveness [22–26]. Inconsistent protective success of *P. gigantea* in different forest types can be partly explained by genetic differences among tree species, which affects susceptibility of wood to infection. In addition, biological control success is affected by stump and root wood conditions during the time of colonization by both the control agent and pathogen, as well differences in the genetic variability and virulence of pathogenic isolates. It should also be noted that there are differences in effectiveness against *Heterobasidion* species from Europe and Northern America. [27,28].

While spectacular protective success using *P. gigantea* is evidenced by rapid reductions in disease development and therefore reduced wood decay in pine stands, in the case of spruce, the preventive and therapeutic effects of treatment of stumps are not always satisfactory [13,24,29–31]. These results are due to many factors, such as differences in the chemical structure of pine and spruce wood, and the pathogen species and strains properties, including differing enzyme activity. However, they are predominantly due to different mechanisms of infection and colonization of roots and trunks. The mechanisms and course of activity of artificially introduced *P. gigantea* have not been fully explained and are still unclear. Successful protection is probably related to the time of infection by competing fungi, the place of the infection in the tree (root or stump) and the interaction between the enzymatically weak pathogen and the enzymatically more active competitor. Małecka et al. [32], Gunulf et al. [33] and Kenigsvalde et al. [34] emphasize the importance of these issues. Much research confirms both the high infectious variability of individual pathogen isolates and variation in the timing of saprotrophic colonization of preserved wood of roots or stumps [13,35]. Sierota et al. [36] described genetic variation among commercial strains and indigenous *P. gigantea* isolates using the random amplified microsatellite (RAMS) method. Several isolates of both competitors did not display significantly different ligninolytic activity during Norway spruce decay. On the other hand, Sierota et al. [37] concluded that isolates with similar dynamics in enzyme activity, even those that are genetically close, may differ in wood decay ability. Sierota et al. [18] suggested that wood density in the tree stump could play a crucial role in the process of infection, in particular the amount of early wood in the annual ring.

The present paper compares hypotheses on factors limiting the effectiveness of the biological control of *Heterobasidion* root rot, in terms of effects on competitors, as well as the roles of environmental conditions. This review also reexamines data from original research performed by the Żółciak and Sierota research group in recent years in Poland.

## 2. Determinants of the Effectiveness of Protective Treatment

### 2.1. Competition as a Dynamic Process of Ecological Stability

Competitive interactions among species have been discussed for many years. The first experiments were performed by Gause in the 1930s [38]. He showed that two species (*Paramecium* spp. and yeasts) occupying the same niche cannot live in stable harmony together, but they can interact in two ways—by dividing the niche or by competitive exclusion. Competitive exclusion can cause local extinction of one of the species. Studies of competitive exclusion using two similar species can give clear results, used by Rishbeth [39] to formulate a biological method of reducing root rot *Fomes annosus* (= *H. annosum*) by a competing fungus *Peniophora* = *P. gigantea*. However, French et al. [40] argued that many common ecological interactions, particularly those involving competition and parasitism, can be

easily confused and that there is often a lack of empirical evidence for reciprocal interactions among species. A similar opinion was expressed by Keddy [41] and Boddy [42]. They underline the distinction between interference competition and competition with exploitation. The nature of competitive and exploitive competition among wood decay fungi is not fully understood [43]. The several types of interactions between competing mycelia were illustrated in Boddy et al. [44]. The authors underline the different value of defensive and aggressive mechanisms in different environments (stressful or un-stressful) and the importance of the sequence of fungal establishment in wood. Observations made in situ show that in nature, there is a complex array of environmental and substrate interactions, such that the relationship between two organisms is highly dependent on these other interactions in addition to direct species-to-species interactions. Maynard et al. [45] showed that in a community rich in species and individuals, competitive exclusion does not occur, and that species richness is a self-reinforcing buffer against strong antagonistic interactions. Furthermore, Maynard [46] suggests also that biotic interactions are much more important than abiotic conditions for explaining fungal interactions. There are species interactions which have significant positive and negative effects on wood-decay fungal activity. Functional outcomes and community structure are largely unrelated to abiotic conditions, but taxonomic richness, evenness, and species associations (i.e., co-occurrence patterns) exhibit strong relationships with community function, which affects decomposition rates and fungal activity. A similar pattern was found by Terhonen et al. [47], who studied the composition of tree endophytes. Accounting for wood traits, fungal species were much more likely to have positive than negative co-occurrence patterns and competitive exclusion was extremely rare, whereas positive interactions among fungal endophytes were more common than expected. Evidence suggested that, across a wide range of wood traits, cooperation may outweigh competition for these fungi. These results shed light on why an artificially introduced competitor sometimes is ineffective in reducing the target fungal disease species.

## 2.2. Influence of Evolutionary History

Knowledge of the molecular mechanisms underlying the interaction between *Heterobasidion* spp. and the biocontrol fungus, *P. gigantea*, is unfortunately still insufficient. There is no evidence of a host genotype or tree species with full resistance to *Heterobasidion* spp. [4]. According to Duxbury et al. [48], who studied the phenomenon of host-pathogen coevolution, genetic variation in susceptibility to infection in natural populations increases as a result of selection by pathogens. On the other hand, genetic variation of pathogens also increases, so that virulence adapts to host population genetics. The most recent ancestor of the genus *Heterobasidion* is estimated to have appeared approximately 160 mya, the same time that gymnosperms differentiated. The ancestral split into two *Heterobasidion* groups, "Pine" (*H. annosum/H. irregulare*, Garbel. and Otrosina) and "Spruce" (*H. parviporum/H. abietinum*, Niemelä and Korhonen, *H.occidentale*, Otrosina and Garbel), occurred approximately 75–85 mya [49].

Studies on the genetic diversity of *H. parviporum* and *H. annosum* have shown that these species have generally low intraspecies genetic diversity [50]. Lower genetic stability of *H. parviporum* was explained as a result of greater gene flow or a recent bottleneck incident. More interestingly, a significant increase in genetic variability was observed within geographically isolated populations of a given *Heterobasidion* species [50,51]. In research concerning the genetic diversity of *H. abietinum*, Zamponi et al. [52] and Luchi et al. [53] found similar results. According to that research, *Heterobasidion* species evolution, and hence genetic diversity, is strongly related to the history of postglacial redistribution from fragmented populations and isolated refugia of their main host species [53]. During the ice age, Norway spruce persisted in refugia, in southern Russia, in the Balkans and to a small degree in southern Italy [54,55]. Holocene recolonization of spruce, and also probably *H. parviporum*, resulted in formation of three main domains: the Baltico-Nordic, the Hercyno-Carpathian and the Alpine [56]. The vast majority of spruce in southern Poland comes from the Alpine and Hercyno-Carpathian refugia, while northeastern Poland and the Scandinavian peninsula are covered mostly by the Baltico-Nordic population. Recolonization of spruce in Europe took place much later than pine, as shown by pollen

records of spruce in Scandinavia that are ca. 2000 years old, while Scots pine (*Pinus sylvestris* L.) was present in the area for almost 9000 years [57,58]. This is reflected in the levels of genetic diversity for *H. annosum* s.s. (population in equilibrium) and *H. parviporum* (lower genetic stability).

The significant interspecific and intraspecific genetic diversity of *Heterobasidion*, resulting from host evolution and glaciation/postglaciation events, may help explain why, despite the high genetic diversity of *P. gigantea* [13,36,37,59], pathogen biocontrol may not always be successful.

## 2.3. Substrate Influence

It is well known that the manner by which *Heterobasidion* pathogen mycelium penetrate root and trunk tissues (Figure 3) is different for pine and spruce [6,8]. Generally, in pine, mycelium of *H. annosum* develops primarily in and around cambial tissues and in the external sapwood, while in spruce, *H. parviporum* mycelium develops in the heartwood [60,61]. While in pines, infected trees die quite quickly due to fungal cells that cause decay of tree tissues at the root collar, in spruce the disease process can last for decades, with infected trees not displaying signs of crown dieback because the cambium and phloem remain healthy and retain their conductive abilities [8]. However, if the spruce are growing in alkaline soils or in the case of high inoclulum pressure, *Heterrobasidion* will still kill them.

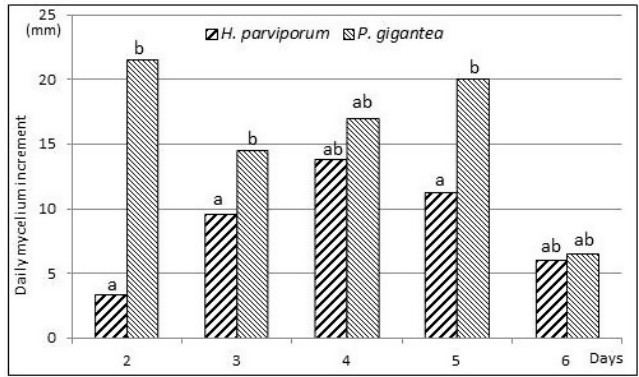

**Figure 3.** Average daily elongation of *H. parviporum* and *Phlebiopsis gigantea* mycelium in the first 6 days of growth in Petri dishes (based on Żółciak et al. [62]). The different letters (a, b) above the bars indicate statistically different groups (significance level at *p*-value < 0.05). Share of the same letter between groups (bars) indicates lack of statistical significance.

The linear growth rate of *H. parviporum* mycelium and of the competitive fungus, *P. gigantea*, evaluated in pure cultures showed an interesting pattern. Żółciak et al. [62] found that, in the first 2 days, differences in growth between both fungi were very significant (Figure 3), with *P. gigantea* elongating about five times faster than *H. parviporum* on 2% malt extract agar (MEA).

After 4 days, mycelial elongation by the competitor significantly accelerated, only to decline after 6 days. At the same time, the mycelium of *H. parviporum* on the same substrate grew much more slowly for much of the time during the first five days, but by day 6 elongation rates were similar. Similar values are described by Gonthier et al. [35], when comparing *H. annosum* and *H. irregulare* mycelial growth in vitro. The comparison of growth rates between several isolates of both fungi showed that mycelial growth rate of *P. gigantea* (FI from Rotstop and GB from PgSuspension) is higher, but elongation rates of some isolates of *H. parviporum* (HP1 and HP3) do not differ significantly from the competitor (Figure 4). The sequences of both partners were deposited in GenBank as: HP1 KX2896987, HP2 KX289698, HP3 KU645328, FI KX756646, and GB KX756647.

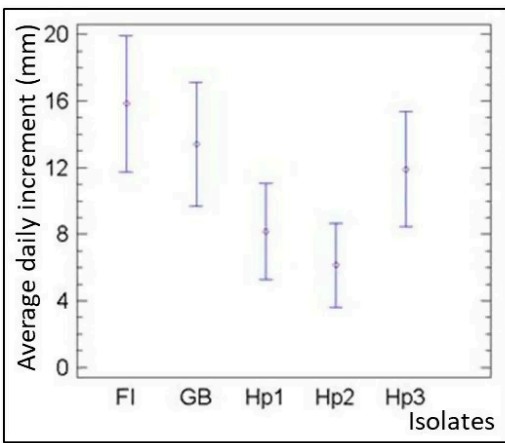

**Figure 4.** Average daily mycelium elongation of *P. gigantea* isolates: FI = Finnish, GB = British; Hp1, Hp2, Hp3 = Polish *H. parviporum* isolates (by Żółciak et al. [62]).

Faster growth of *P. gigantea* mycelium (Figure 5) results not only from its genetically determined ability to degrade host plant cell walls, but also from the structure of the wood itself. Tomczak and Jelonek [63] reported that the physical and chemical structure of the cell walls in Scots pine growing in previously fertilized (post-agricultural) soils is definitely different, with lower wood density, especially in the peripheral part of the tree, compared to pines growing in forest soils.

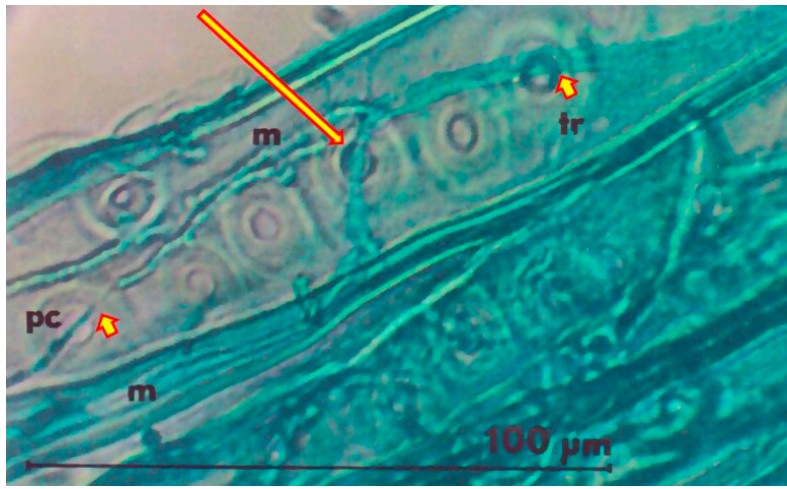

**Figure 5.** *Phlebiopsis gigantea* hyphae penetrating pine sapwood (cross-section: m = mycelium, tr = tracheid, pc = pit cells; arrows indicate hyphae) (Sierota [64]).

Different *P. gigantea* isolates show not only different rates of growth of mycelium on agar media in the laboratory, but also in the rate of decay of root wood in the forest. What could be the reasons for such fluctuations in growth rate of mycelium biomass? The answer may be found by examining the rate and thinning of the production of enzymes secreted by both fungi.

## 2.4. Enzyme Activity

Adomas et al. [65] showed a high differentiation in the competitive activity of both *Heterobasidion* spp. and *P. gigantea*, as indicated by mRNA expression. Fungi such as *P. gigantea* decompose freshly cut stumps and roots by the production of extracellular enzymes that degrade the lignin-cellulose complex. Lignins are decomposed primarily by oxidoreductases, such as lignin peroxidase, manganese peroxidase and laccase, and cellulose by hydrolases, such as cellulases [66]. Because *P. gigantea* is also able to degrade resinous extractives, it is characterized as a pioneer colonizer [67]. During

lignin decomposition, phenolic compounds are polymerized and co-polymerized. The intensity of these processes can be measured by the amount of respiratory enzymes such as dehydrogenases and phosphatases [68–70]. The production and activity of some enzymes involved in the infection process and the decay of wood tissues was described for *Heterobasidion* spp. by Maijala et al. [71], Johansson et al. [72], and Maijala [73], and for *P. gigantea* by Schafer et al. [74–76].

Żółciak et al. [26] evaluated metabolites of *P. gigantea* in Norway spruce wood and found a strong correlation between phosphatase and dehydrogenase concentrations, which were negatively correlated with the concentration of cellulase. Peroxidase production was not correlated with cellulase production, but was strongly correlated with the presence of vanillic and protocatechuic acid. Xu et al. [77] describe pathways of enzymatic degradation of lignin, where protocatechuic and vanillic acids are key byproducts of decay. Żółciak et al. [26] also found that the concentration of protocatechuic acid was many times greater than that of vanillic acid. Protocatechuic acid is the common byproduct of two pathways of lignin degradation, namely G-lignin and H-lignin decay, whereas vanillic acid appears only during G-lignin decomposition. Varela and Tien [78] described similar relationships, and attributed them to active respiratory hydroxylation of p-hydroxybenzoic acids in wood. The concentrations of both cellulases and peroxidases produced by different isolates of *P. gigantea* in decayed Norway spruce wood were not significantly different. Average cellulase concentration for Finnish isolates of *P. gigantea* was 11.89 μg of reduced sugars/g dry weight, and 11.26 μg/g for British isolates, whereas average peroxidase was 0.99, and 1.03 U/g dry weight, respectively [36,37].

Phenolic acids induced by *P. gigantea* and *H. annosum* participate in fungal development and play important roles in the processes of tree resistance to fungal infection and the decay of wood cell walls [79]. Sierota et al. [80] found an increase in some phenolic acids during fungal attack of Scots pine of different provenances, with significant positive correlations between the amounts of hydroxybenzoic with vanillic and cinnamic acids, cinnamic with o-coumaric acids, and negative correlations with the amounts of ferulic and salicylic acids in the phloem cells. Mukherjee and Kundu [81] described the inhibitory effect of benzoic acid on some fungi, which was confirmed by Haars and Hüttermann [82].

The data presented in Table 1 indicate key differences between *P. gigantea* and *H. annosum* in competition for resources, resulting from divergent enzymatic pathways. This is indicated by the absence of laccases shown by *P. gigantea* isolates, and the high activity of this enzyme in the fruiting bodies of *H. annosum* (Table 1). Laccase is the basic enzymatic indicator of the pathogenicity of white-rot fungi [83,84]. The activity of other enzymes evaluated was at similar levels in *P.gigantea* and *H. annosum*. Peroxidase activity in both *P. gigantea* and *H. annosum* mycelia indicates the activation of lignin biodegradation [68,69]. For the pathogen, phosphatase activity in wood was slightly more than for *P. gigantea*. Phosphatase is important in phosphorus uptake and is utilized after wood hydrolysis, [70] coinciding with the activity of cellulase. For *P. gigantea* isolates, the activity of phosphatase was stimulated to varying degrees by dehydrogenase. Interesting differences were found in peroxidase, cellulase, and dehydrogenase activity between *P. gigantea* isolates. Mycelium of a *P. gigantea* isolate collected from decaying pine stump wood showed significantly less peroxidase and cellulase activity than mycelium taken from the fruiting body. The activity of peroxidase and cellulase in the mycelium obtained from the sporocarps of *P. gigantea* and *H. annosum* was highly similar. Samils et al. [85] observed that not all *P. gigantea* isolates could be treated as active saprotrophs and may show weak antagonistic effects, suggesting that resistance towards the competitor increased over time. Mgbeahuruike et al. [25] compared two hydrophobin *P. gigantea* genes. This flexible property is also indicated by the characteristics of color change of guaiacol medium in reaction to phenol oxidase [86], illustrated in Figure 6 (right), where isolate 1 is more saprotrophic, and isolate 2 is less antagonistic.

**Table 1.** Average enzymatic activity of isolates of *P. gigantea* and *H. annosum* mycelium growing on Scots pine wood, calculated on dry wood mass basis (data modified from Żółciak et al. [76]).

| Isolate Origin | Enzyme Activity | | | | |
| --- | --- | --- | --- | --- | --- |
| | Laccases U $g^{-1}$ | Peroxidases U $g^{-1}$ | Cellulases µg glucose $g^{-1}$ | Phosphatases µg p-PNP $g^{-1}$ | Dehydrogenases mg TPF $g^{-1}$ 24 $h^{-1}$ |
| P.g.$_5$ stump | 0 | 0.608 | 688.86 | 295.11 | 51.42 |
| P.g.$_6$ fruiting body | 0 | 8.802 | 3280.72 | 300.34 | 25.15 |
| H.a. fruiting body | 1482.785 | 7.296 | 4269.08 | 351.88 | 58.12 |

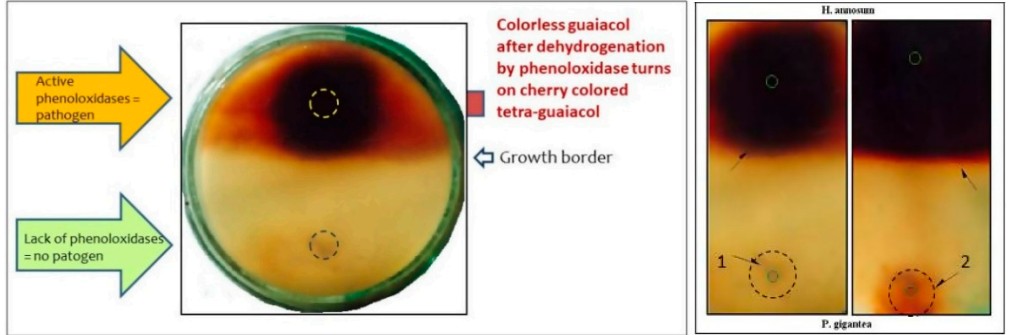

**Figure 6.** Color reaction of pathogen (*H. annosum*) and saprotroph (*P. gigantea*) enzymes in dual cultures on agar medium with quaiacol (left) and comparison of enzymatic activity of two *P. gigantea* isolates (right), 1 and 2—*P. gigantea* isolates, (Sierota [87]).

## 2.5. Wood Decay

Wood decomposition by fungi is rather fast [88,89]. Fackler et al. [90] described rapid lignin degradation by some white-rot fungi, occurring over a matter of days. Wood decay rate significantly depends on the origin of the culture. For *P. gigantea*, differences between mycelium isolated from stump wood or a fruiting body of forest origin, and a pure culture from commercial Rotstop, were described by Żółciak et al. [62]. Mgbeahuruike et al. [91] did not find differences in the decay rate of Norway spruce and Scots pine wood, but decay rates for Rotstop isolates were different. Sierota [64] found dry wood weight loss in Scots pine stumps after 6 months from silvicultural harvesting of 2.95%, 17.46%, and 21.84% for stump wood collected from salvage cutting, routine cutting and in stumps inoculated with *P. gigantea*, respectively. At the same time, the average decay of lateral roots was 39.4%, with roots that were in moist conditions (after water saturation) being more decayed than from drier conditions (no saturation).

Wood decay activity of *Heterobasidion* and *Phlebiopsis* determines the success of biological control, especially since wood decomposition can be also slowed down by the bordering of xylem defense wood [92]. Żółciak et al. [62] compared the rate of Norway spruce decay between isolates of *Heterobasidion* and *Phlebiopsis* and found ~33% wood loss for *P. gigantea* isolates and ~25% loss for *H. parviporum* isolates (Figure 7).

Differences in the specific gravity of wood can indicate the ratio between spring and late wood [93,94] which is correlated with mycelial growth rates in tissues and therefore to the degree of root decay by both *Heterobasidion* and *Phlebiopsis*. Sierota [64] described the rapid growth of *P. gigantea* and commensurate Scots pine wood decay in roots with high specific gravity. Sierota et al. [18,19] found the same results for the decay of Norway spruce wood by different *H. parviporum* isolates.

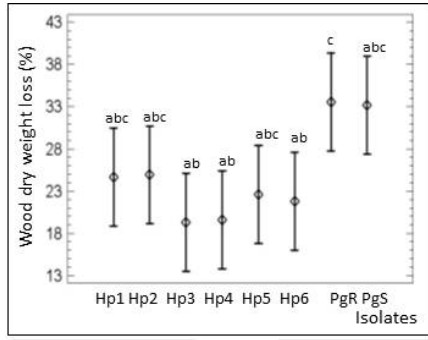

**Figure 7.** Average loss (%) of dry weight from Norway spruce wood after 6 months of decomposition, caused by six *H. parviporum* and two Finnish *P. gigantea* isolates (Tukey test, $p_\alpha = 0.05$) (Żółciak et al. [62]). The different letters (a, b, c) above the bars indicate statistically different groups (significance level at *p*-value < 0.05). Share of the same letter between groups (bars) indicates lack of statistical significance.

### 2.6. Impact of Abiotic Conditions

Soil temperature and precipitation are directly and/or indirectly correlated with the occurrence of root rot. Korhonen and Stenlid [95] reported that the optimal temperature for mycelium growth of *H. annosum* is ~22 °C, which can produce mycelium elongation in the laboratory of as much as 9 mm per day. Cooding et al. [96] showed that in hot years, when air temperature exceeds 35 °C, and in years with severe cold [97], the infection of stumps by pathogen spores is reduced. Mykhayliv and Sierota [98], in analyzing the presence of the pathogen in Polish forests between 1975 and 2007, found a strong relationship between weather conditions in particular months in year *n* − 1 and the area of stands affected by *Heterobasidion* spp. in year *n*. The impact of weather on the presence of disease symptoms was stronger in stands older than 20 years.

The authors concluded that disease development can be stimulated by higher soil temperature at the end of the growing season and at the beginning of winter. Witzell et al. [99], however, did not find a simple relationship between temperature regime and the occurrence of *H. annosum* and *H. parviporum*. Furthermore, both pathogen, competitor, and interactions between them can be suppressed by changing $O_2$ and $CO_2$ concentrations in air and in wood. Kern and Linkies [100] found in vitro lower expression of *H. parviporum* hydrophobin genes (HAH1 and HAH2) with $O_2$ reduction to 0.4%.

### 2.7. Impact of Biotic Interactions

*Heterobasidion* and *Phlebiopsis* fungi are components of the broader ecosystem and should therefore be evaluated in a way that takes into account the impact of the rhizosphere and endophytic biota in roots on both fungi. Vasiliauskas et al. [101] suggest that there is a long-term positive effect of *P. gigantea* in Rotstop on mycobiota in Norway spruce stumps. Primary and secondary infections from the soil affect *Heterobasidion* through the actions of soil microorganisms, including fungi and bacteria. However, infection of stump and root wood tissues by *Heterobasidion* and *Phlebiopsis* after tree harvesting is affected by the biotic environment in a similar way. The interesting mutual relations of microorganisms inhabiting roots and stumps artificially infected with *P. gigantea* was the object of the present authors' interest [18,19,26,102–105]. Results indicated that the presence of the active wood decomposer (*P. gigantea*) in the wood of Norway spruce stumps did not deplete the richness of microorganisms that inhabit stumps.

The other factor influencing fungi vitality in wood is the presence, composition, and structure of the mycetophagous guild of invertebrates, including mites, nematodes, springtails, insects and molluscs. Some studies showed the role of invertebrates (e.g., *Hystiogaster* spp., *Tarsonemus* spp.) in disseminating fungal spores. Mercado [106] indicates that some bark beetles and their symbiotic mites are fungus feeders, but others are known to feed on nematodes, other mites, bark beetle eggs, and larvae. The understanding of such ecological networks is far from completely resolved [107–109]. The results

obtained by our team suggest that some crystal-forming fungi are avoided by invertebrates [103]. Nagy et al. [92] discuss the enhanced production of oxalic acid by *H. parviporum* as a response to the tree defense system. Nagy et al. [92] conclude that trees induce the formation of reaction zones that possess antimicrobial properties, such as elevated pH and cation content; pathogens lower substrate pH by secreting oxalic acid, with its conjugate base oxalate being a reducing agent t as well as a chelating agent for cations. On the other hand, Prasad and Shivay [110] showed that both oxalic acid and calcium oxalate possess plant properties against wood-consuming organisms—both insect pests and grazing animals. We are convinced that this mechanism is also involved in fungal resistance to invertebrates, as suggested by Binns [111] and White [112]. Calcium oxalate, whether produced by fungi or applied externally, has been shown in mushroom production systems to protect fungi from feeding damage by sciarid flies. The effects of protective chemical compounds on the presence and composition of wood-consuming organisms could strongly influence competition between *P. gigantea* and *Heterobasidion* spp.

*2.8. Timing P. gigantea Treatment*

Efficacy of treatment with *P. gigantea* in the forest has been investigated many times. Kenigsvalde et al. [34] reported as much as 20% greater efficacy in conifer stands of Latvian isolates of the competitor compared to the commercial formulation of Rotstop. Małecka et al. [102] and Żółciak et al. [17] also found significant differences between isolates when comparing decay in Scots pine and Norway spruce stumps of twelve unregistered Polish isolates with two Finnish (Verdera), and four British (PgSuspension) isolates. Differences were found not only among isolates from the same geographical origin or sampled at different times during the process of decay, but also according to the place of isolation—whether from the soil spore bank, root, or stump, as well as according to whether treatment was applied in the spring or autumn. Similar observations were presented by Vainio et al. [13], Berglund and Rönnberg [22], Annesi et al. [23], Nicolotti and Gonthier [11], Żółciak et al. [76] and Drenkhan et al. [113].

Sierota [64] found that roots and stumps with higher moisture content, e.g., from salvage-cut trees, or those inoculated in the rainy season with *P. gigantea*, can be effectively overgrown by the saprotroph, similar to freshly cut stumps of recently harvested trees in commercial thinnings. The results obtained in many studies show a significant impact of stump and root wood moisture depending on the humidity during the period of treatment (spring or autumn) on the success of *P. gigantea* treatment [32,64,101,104]. These results indicate that biological treatment could be more effective when it is carried out in weather conditions favorable for fungus growth, which rules out treatment during dry summer weather.

## 3. Perspectives

Published information and field studies indicate that rates of mycelium development of some *Heterobasidion* isolates, both in pure culture and from roots in the forest (secondary infections), can be comparable to mycelium development by isolates of *P. gigantea*. However, *P. gigantea* isolates caused greater decomposition of conifer wood than most, and in some cases all, pathogen isolates that were evaluated. It should be presumed that similar relationships occur in infected wood of roots and stumps in natural conditions, even if they have not been directly observed. Such a phenomenon may be common in forest stands established on post-agricultural land or rich forest sites, on which trees produce wood with larger annual rings than in trees on poorer forest soils [63]. It has been confirmed that a fast-growing pathogen mycelium, growing from the roots towards the stump, can penetrate wood tissues just as fast as or faster than the mycelium of a weak growing competitor introduced into the cut stump. The danger is that the mycelium of the pathogen can infect the roots some years earlier and colonize the stump faster than mycelium of saprotrophs introduced at a later time into the stump. Under this scenario, the pathogen can effectively compete with *P. gigantea*, applied in artificial inoculation to the stump's surface. It seems that this is one of the main factors limiting the effective colonization of spruce stumps by *P. gigantea* in treating areas of *Heterobasidion* root rot occurrence.

As Greig and Pratt mentioned [114], *Heterobasidon* spp. have been found in stumps 62 years after felling and in root systems of diseased trees for several decades after harvesting.

Greater chances of successful treatment occur when there is a program in place to monitor phytopathological threats and when protective treatment is carried out at the earliest time. Boddy et al. [44] advises additional silviculture practices to fight against pathogens. The first practice is delaying thinning until trees are older and to conduct thinning operations when basidiospores are not being dispersed [44]. Another practice is to facilitate the rapid and inexpensive registration of isolates of *P. gigantea* that have been found to be most effective in a given environment/country. It is not helpful when regulations prevent the replacement of an isolate registered in a commercial preparation for years with a more active local isolate of the same species, since the ability to carry out such substitutions protects the forest from root rot.

A more fundamental question is what a 'healthy amount of the disease' is in a commercial forest [115]. Even when large-scale biotic threats like root and butt rot are present, ecosystems are able to compensate for lost elements and to recreate them in a new, often better (though certainly different) form. The structural components of forests can be maintained only for relatively short intervals, not only due to regulatory constraints, but also the natural capacity of the ecosystem to restore lost or distorted trophic structures, as well as due to human intervention [116]. The open question is whether to "restore" the distorted ecosystem to its previous form, or to skillfully respond through natural ecosystem processes. In healthy natural ecosystems, pathogens, predation, and other natural damage factors, including those sometimes occurring at large scales, are phenomena that are often desirable as they allow restorative forest processes to act [117–119]. In managed stands, the activity of pathogens can be controlled, for example, by avoiding threats, applying the principle of risk mitigation, or by semi-natural forest breeding.

## 4. Conclusions

Further research is needed on approaches for identifying threats from pathogens, on the properties of individual isolates of *P. gigantea* (new formulations and isolates) and *H. parviporum*, on the timing of protective measures, and on the search for other species of saprotrophic fungi that effectively decompose stump wood, especially those that occur in spruce. The use of biological control methods that employ natural competitors to restrict, rather than to eliminate, *Heterobasidion* spp. occurrence, seems to be the best approach to protecting the forest.

**Author Contributions:** The manuscript was written by all authors. The general conception of the project was provided by Z.S. All authors have read and agreed to the published version of the manuscript.

**Funding:** This work was partially supported by the State Forest Holding in Poland through Project No. 500 426, and by statutory funds of the Forest Research Institute in Sękocin Stary, the Warmia and Mazury University in Olsztyn and the University of Warsaw.

**Acknowledgments:** The authors would like to thank the anonymous reviewers for their valuable comments and suggestions.

**Conflicts of Interest:** Authors declare there were no personal circumstances or interest that may be perceived as inappropriately influencing the representation or interpretation of reported research results.

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
