# Peer review of "Why Does Phlebiopsis gigantea not Always Inhibit Root and Butt Rot in Conifers?"

_forests, doi:10.3390/f11020129_

Round 1

Reviewer 1 Report

This review focuses on the effectiveness of the biological control agent Phlebiopsis gigantea in use against important root and butt rot agents of conifers in the genus Heterobasidion. The work is interesting, but prior to accepting multiple modifications and changes are required to clarify certain points and correct a few minor errors. 

Line 27 (in the abstract): the word 'biocenosis is accurate, but not in particularly common usage in plant pathology. Please say what you mean.

Line 29: delete 'disease'.

Line 35: why are you using 'fungi', then mention one species (even if that out-of-date 'species' was actually a complex)?

Line 35: continuing from the above point, it is now over 20 years since the European Heterobasidion species were properly defined. Using the 'sensu lato' species name is incorrect.

Line 43: what are 'factors of pathogenicity'?

Line 50: 'the aforementioned' is clumsy English.

Line 65: I have always had my doubts about the relevance of these observations, made in entirely artificial, laboratory-based systems, to what actually happens in forests.

Line 68: 'This' has no object. This what?

Line 70: the word 'resistance' would refer to host-pathogen interactions. In no sense of the word is P. gigantea a pathogen; it is a saprotrophic decay fungus.

Line 74: 'effectiveness of Heterobasidion'? I think you mean 'against'.

Line 80: delete 'in', in the term 'including differing (in) enzyme activities'.

Line 85: I have no idea what an 'enzymatically weak pathogen' or 'enzymatically more active competitor' are. If the organisms were 'enzymatically weak', they would not survive.

Line 89: Why is 'preserved' here? It is not relevant to the system being discussed.

Line 111: use the past tense.

Line 114: delete 'several'. Also use the past tense in this sentence.

Line 120: the short sentence 'This problem was studied by...' is not required - delete it.

Line 121: Maynard did not work alone! Maynard et al. [also in line 123].

Line 129: past tense!

Lines 132-135: Freshly cut stumps may have endophytes, but the colonies are not extensive: how would these highly localised 'infections' really influence establishment of P. gigantea added in a huge inoculum dose?

Line 148: Clumsy English. Delete 'the genetic diversity of'.

Lines 156-157: 'found refuge in' implies the tees rushed to that refugium. Maybe better to say' Norway spruce persisted in refigia in southern Russia...'?

Line 171: delete 'pathogen'; make 'penetrate' plrual.

Line 17173: 'develops'.

Line 174: [develops again]; 'the middle part of the tree' is vague - just heartwood.

Line 175: all living fungal cells are 'enzymatic', otherwise they would not be alive! Use 'fungal enzymes'.

Lines 175-178: if the spruce are growing in alkaline soils, or there is high inoculum pressure, Heterobasidion will still kill them.

Line 184: State the culture medium used for this work - it has a HUGE effect on interaction outcomes.

Line 187: delete 'in elongation rate'.

Line 202: you have used a strange way to express this particular point here. Delete 'enzymatically break down', and replace with 'degrade'.

Line 205: why 'definitely'? If it is certain, then there must be references to support your assertion.

Line 211: 'seuqence' is confusing here. I think you mean timing of production; sequence could refer to the amino acid sequence.

Line 214: delete 'fungi'.

Line 216: laccase is only one enzyme involved here. It is a complex of oxidoreductase enzymes that degrades lignin, and is probably mainly driven by manganese peroxidases.

Line Lines 219-220: replace 'the amount of' with 'quantifying'.

Line 231: 'This' has no object.

Lines 245-246: you use 'aforementioned' again. Please be specific.

Line 248: Cweilong and Huttermann may have used H. annosum, but it is not certain, as their work predated full descriptions of the European species in the complex.

Line 252: why 'causing'. The 'resulting from'.

Line 253: what does 'its' refer to?

Line 254: are most white rot fungi pathogens? No, they're not, in the true sense. They are highly specialised saprotrophs.

Line 260: I am not sure how the effects of dehydrogenase on phosphatase activity would be tested.

Line 266: what does 'resistance' mean here?

Figure 6: the phrase beginning 'colourless guaicol' does not make sense. It might be fixed by replacing 'on' with 'to'. The caption to this figure is also inadequeate. It does not explain what the image in the box on the right shows.

Line 277: Replace 'culture' with 'isolate'.

Line 284: what does 'moist conditions' mean? How 'moist' were the conditions. Maybe the soil type should be mentioned?

Line 287: what do you mean by 'boredering of xylem defence wood'? It does not make sense.

Line 296: 'Differences'.

Line 297: replace' related' with 'correlated'.

Line 300: the 'same' result as what?

Line 304: growth of which mycelium? You need to specify which fungus you are talking about.

Line 310: the area of what affected by Heterobasidion?

Lines 316-318: these measurements of gene expression were made in vitro. It is very important to state that fact.

Line 327: 'interesting' is a vague and personal value judgement. I suggest simply deleting 'interesting'.

Line 330: 'biota'. Please use more careful wording to explain what you really mean.

Line 334: 'ex.'? Would 'e.g.' be more appropriate?

Line 335: past tense.

Line 343: 'reductant' is not the correct word.

Line 345: Plants do not practice 'self-defence'. Use more appropriate wording.

Line 360: What is the 'mycelium bank'? Use correct terminology.

Line 388: It is Boddy et al.! I am sure this problem occurs at several points in the manuscript. Correct throughout!

Line 389: The discovery of Heterobasidion still alive in larch stumps 62 (or was it 63) years after felling what not the work on Boddy et al.! Use the correct reference.

Line 395: 'quick and cheap'. Better to say rapid and inexpensive, to avoid ambiguities.

Lines 400-407: I cannot see the relevance of this paragraph. I suggest that you delete it all.

Line 410: 'them'? What do you mean?

Line 415: why 'exactly'? It does not make sense.

Acknowledgements: How could AZ and KS generate 'data' - the manuscript is a review.

References are not formatted consistently.

Author Response

Answer to Reviewer 1

Thanks for the valuable comments and suggestions which have led to significant improvements to the quality of this paper. We tried to meet the expectations of the reviewers, but some suggested changes could significantly increase the size of the text. In what follows, we shall detail the changes we have made in the paper.

The text was improved and corrected by plant biologist – Englishman, who has been working as a plant pathologist for many years and has been the editor of plant biology magazine for many years. Therefore we had agreed that the native English is correct; however, “nobody is perfect”.

We agree with Reviewer 1 statements, and the newly submitted manuscript is correct according to the advice, and in details as below.

 Line cited by reviewer

Reviewer suggestion

Authors answer

Line 27

(in the abstract): the word 'biocenosis is accurate, but not in particularly common usage in plant pathology. Please say what you mean.

We have changed line 27 to” …and the richness of the fungal biota…

It’s more precise in the context of the data presented in References 103, 106 or 107

Line 29

delete 'disease'.

We have deleted this word

Line 35

why are you using 'fungi', then mention one species (even if that out-of-date 'species' was actually a complex)? continuing from the above point, it is now over 20 years since the European Heterobasidion species were properly defined. Using the 'sensu lato' species name is incorrect.

We agree with you

We have corrected the text according to this suggestion Heterobasidion spp.

Line 43

what are 'factors of pathogenicity'?

For example: genetic variability, enzymatic activity, antifungal ability, coexistence with other fungi and bacteria, etc.

Line 50

“the aforementioned' is clumsy English.

We have changed to “ …competitor P. gigantea, remains…

Line 65

I have always had my doubts about the relevance of these observations, made in entirely artificial, laboratory-based systems, to what actually happens in forests.

The sentence based mainly on References 20, 21.

Line 68

'This' has no object. This what?

“These limitations…” added

Line 70

the word 'resistance' would refer to host-pathogen interactions. In no sense of the word is P. gigantea a pathogen; it is a saprotrophic decay fungus.

We changed to “susceptibility of wood”..

Line 74

'effectiveness of Heterobasidion'? I think you mean 'against'.

Yes, thank you

Line 80

delete 'in', in the term 'including differing (in) enzyme activities'

Has been deleted

Line 85

I have no idea what an 'enzymatically weak pathogen' or 'enzymatically more active competitor' are. If the organisms were 'enzymatically weak', they would not survive.

Our concept of “weak/strong pathogen” and “ weak/strong saprotroph” is based on some References and own research. In many laboratory tests with guaiacol as tester for phenoloxydase  activity, as well as in the tests of mycelium growth in dual cultures  we noted  different reactions for different isolates of both fungi.

Line 89

Why is 'preserved' here? It is not relevant to the system being discussed

We added : “of roots or stumps”

Line 111

use the past tense.

We have used

“argued”

Line 114

delete 'several'. Also use the past tense in this sentence.

Has been done

Line 120

the short sentence 'This problem was studied by...' is not required - delete it.

Has been deleted

Line 121

Maynard did not work alone! Maynard et al. [also in line 123].

Has been added “et al.”

Line 129

past tense!

Has been done

Lines 132-135

Freshly cut stumps may have endophytes, but the colonies are not extensive: how would these highly localised 'infections' really influence establishment of P. gigantea added in a huge inoculum dose?

P. gigantea in ROTSTOP  not always works satisfactory. Our research (Ref. 105-107) confirmed the presence of active fungal biota in roots and stumps before and after the treatment. We rejected this sentence.

Line 148

Clumsy English. Delete 'the genetic diversity of'.

Has been deleted

Lines 156-157

'found refuge in' implies the tees rushed to that refugium. Maybe better to say' Norway spruce persisted in refigia in southern Russia...'?

We have changed

Line 171

delete 'pathogen'; make 'penetrate' plural.

“pathogen” has been deleted

Line 171-173

'develops'.

Has been done

Line 174

[develops again]; 'the middle part of the tree' is vague - just heartwood.

Has been done

Line 175

all living fungal cells are 'enzymatic', otherwise they would not be alive! Use 'fungal enzymes'.

Has been done

Lines 175-178

if the spruce are growing in alkaline soils, or there is high inoculum pressure, Heterobasidion will still kill them.

We have added sentence:

“However….

Line 184

State the culture medium used for this work - it has a HUGE effect on interaction outcomes

We have added:

“..on 2% malt extract agar (MEA).

Line 187

delete 'in elongation rate'.

Has been done

Line 202

you have used a strange way to express this particular point here. Delete 'enzymatically break down', and replace with 'degrade'.

We have deleted

Line 205

why 'definitely'? If it is certain, then there must be references to support your assertion.

We have deleted

Line 211

'seuqence' is confusing here. I think you mean timing of production; sequence could refer to the amino acid sequence.

We have changed

Line 214

delete 'fungi'.

We have deleted

Line 216

: laccase is only one enzyme involved here. It is a complex of oxidoreductase enzymes that degrades lignin, and is probably mainly driven by manganese peroxidases.

We have added:

“lignin peroxidase, manganese peroxidase…”

Lines 219-220

replace 'the amount of' with 'quantifying'.

Has been done

Line 231

'This' has no object.

We have changed

Lines 245-246

you use 'aforementioned' again. Please be specific.

deleted

Line 248

Cweilong and Huttermann may have used H. annosum, but it is not certain, as their work predated full descriptions of the European species in the complex.

The sentence has been shortened.

Line 252

why 'causing'. The 'resulting from'.

changed

Line 253

what does 'its' refer to?

To “this enzyme”; it has been changed

Line 254

are most white rot fungi pathogens? No, they're not, in the true sense. They are highly specialised saprotrophs

Has been changed

Line 260

I am not sure how the effects of dehydrogenase on phosphatase activity would be tested.

We rejected this sentence as not  precise

Line 266

what does 'resistance' mean here?

We replaced the text

Figure 6

: the phrase beginning 'colourless guaicol' does not make sense. It might be fixed by replacing 'on' with 'to'. The caption to this figure is also inadequeate. It does not explain what the image in the box on the right shows.

It has been changed

Line 277

Replace 'culture' with 'isolate'.

Has been done

Line 284

what does 'moist conditions' mean? How 'moist' were the conditions. Maybe the soil type should be mentioned?

The wood was saturated with water [63] We changed the text

Line 287

: what do you mean by 'boredering of xylem defence wood'? It does not make sense

We rejected this word

Line 296

'Differences'.

Has been done

Line 297

replace' related' with 'correlated'.

Has been done

Line 300

the 'same' result as what?

Has been clarified

Line 304

growth of which mycelium? You need to specify which fungus you are talking about.

We have added

Line 310

the area of what affected by Heterobasidion?

We have added

Lines 316-318

these measurements of gene expression were made in vitro. It is very important to state that fact.

We have added

Line 327

'interesting' is a vague and personal value judgement. I suggest simply deleting 'interesting'.

We have deleted this word

Line 330

 'biota'. Please use more careful wording to explain what you really mean.

Has been done

Line 334

'ex.'? Would 'e.g.' be more appropriate?

Has been done

Line 335

past tense.

Has been done

Line 343

 'reductant' is not the correct word.

we changed on ‘reducing agent’

Line 345

Plants do not practice 'self-defence'. Use more appropriate wording.

Has been changed

Line 360

What is the 'mycelium bank'? Use correct terminology.

We changed on ‘soil spore bank’

Line 388

It is Boddy et al.! I am sure this problem occurs at several points in the manuscript. Correct throughout!

Has been done

Line 389

The discovery of Heterobasidion still alive in larch stumps 62 (or was it 63) years after felling what not the work on Boddy et al.! Use the correct reference.

It was changed. The first Reference was by Greig and Pratt 1976 [116]

Line 395

 'quick and cheap'. Better to say rapid and inexpensive, to avoid ambiguities.

Has been done

Lines 400-407

I cannot see the relevance of this paragraph. I suggest that you delete it all.

We have deleted

Line 410

'them'? What do you mean?

deleted

Line 415

why 'exactly'? It does not make sense.

deleted

Acknowledgements:

How could AZ and KS generate 'data' - the manuscript is a review.

generate the outgoing data. It has been rejected

References are not formatted consistently.

Has been done

Reviewer 2 Report

Strong points

The work contributes to clarify the performance of  Phlebiopsis gigantea in different conifers species and provides reliable information about this subject.

Weak points

Despite the importance of the topic the manuscript lacks focus. The work is poorly structured and requires a thorough revision.

Comments

Introduction

Is necessary:

To describe causes that may explain the differences observed between conifers (described in 2). In this section, you can use picture 1 inserted in the text, but do not add any more pictures. 

Suggestion: Move to Introduction everything you have described in point 2 and rewrite this section carefully.

The purpose of the work is thus not clearly specified. Rewrite it clear in the end of Introduction and point out its relevance.  

Create a Material and Methods Section

In this section, organise clearly the information analysed. I suggest present it in a Table.

Create a Results and Discussion Section

Present in this section all the graphics. Analyse the results based in bibliographic references.

Author Response

Answer to Reviewer 2

Many  thanks for the advices, but we cannot to agree with Reviewer’ 2  suggestions to rewrite the paper and present the text as typical original research. The suggestions to rewrite whole  text to the form of typical research paper, with Materials, Results, and Discussion sections, are out of our purpose.

The problem indicated by the Reviewer 2 requires an additional and broader study. In our concept we describe the problems, more or less holistic.

Reviewer 2                                                                     

Introduction

Is necessary: To describe causes that may explain the differences observed between conifers (described in 2). In this section, you can use picture 1 inserted in the text, but do not add any more pictures. 

Suggestion: Move to Introduction everything you have described in point 2 and rewrite this section carefully.

The purpose of the work is thus not clearly specified. Rewrite it clear in the end of Introduction and point out its relevance.  

The presented article is submitted as a review paper, and only in some cases based on original results of the authors team, not published in this form -  we indicated this disclaimer in lines 98-99.

Therefore the proposed suggestions to rewrite this text to the typical research paper, with Materials, Results and Discussion sections, is out of our purpose.

Differences has been explained in line 40 (Fig. 1) [lines 39=42 of new text], lines 75-78, in References 6,8, and in particular Chapters.

Introduction describes general subject matter  of the problem, whereas next Chapters describe particular mechanisms and processes.

The main idea of the paper can be summarized as follows:

Mycelium of some strains of the pathogen can infect the roots some years earlier and colonize the stump faster than less dynamic mycelium of saprotroph introduced at a later time into the stump.

This suppose is discussed based on the literature and the research of the authors.

Create a Material and Methods Section            In this section, organise clearly the information analysed. I suggest present it in a Table.

We consequently do not change the main purpose.

Create a Results and Discussion Section         Present in this section all the graphics. Analyse the results based in bibliographic references.

Round 2

Reviewer 2 Report

The work is much clear  than the previous version.

This new version, more compreensible, justifies the authors' choice of structure and allows not follow the traditional model. 

I suggest that point 2.3 be renamed to Substrate Influence

Author Response

Answer to Reviewer 2

Thanks very much for your comments and suggestion. We renamed point 2.3. to “Substrate influence”. The text was improved and corrected by plant biologist – Englishman, who has been working as a plant pathologist for many years and has been the editor of plant biology magazine for many years.

Best regards

Authors
